# HELLP Syndrome and Differential Diagnosis with Other Thrombotic Microangiopathies in Pregnancy

**DOI:** 10.3390/diagnostics14040352

**Published:** 2024-02-06

**Authors:** Stefano Raffaele Giannubilo, Daniela Marzioni, Giovanni Tossetta, Andrea Ciavattini

**Affiliations:** 1Department of Clinical Sciences, Polytechnic University of Marche, Via Corridoni 11, 60123 Ancona, Italy; a.ciavattini@staff.univpm.it; 2Department of Experimental and Clinical Medicine, Polytechnic University of Marche, Via Tronto 10/a, 60126 Ancona, Italy; d.marzioni@staff.univpm.it (D.M.); g.tossetta@pm.univpm.it (G.T.)

**Keywords:** HELLP, HUS, pregnancy, TMA, TTP

## Abstract

Thrombotic microangiopathies (TMAs) comprise a distinct group of diseases with different manifestations that can occur in both pediatric and adult patients. They can be hereditary or acquired, with subtle onset or a rapidly progressive course, and they are particularly known for their morbidity and mortality. Pregnancy is a high-risk time for the development of several types of thrombotic microangiopathies. The three major syndromes are hemolysis, elevated liver function tests, and low platelets (HELLP); hemolytic uremic syndrome (HUS); and thrombotic thrombocytopenic purpura (TTP). Because of their rarity, clinical information and therapeutic results related to these conditions are often obtained from case reports, small series, registries, and reviews. The collection of individual observations, the evolution of diagnostic laboratories that have identified autoimmune and/or genetic abnormalities using von Willebrand factor post-secretion processing or genetic–functional alterations in the regulation of alternative complement pathways in some of these TMAs, and, most importantly, the introduction of advanced treatments, have enabled the preservation of affected organs and improved survival rates. Although TMAs may show different etiopathogenesis routes, they all show the presence of pathological lesions, which are characterized by endothelial damage and the formation of thrombi rich in platelets at the microvascular level, as a common denominator, and thrombotic damage to microcirculation pathways induces “mechanical” (microangiopathic) hemolytic anemia, the consumption of platelets, and ischemic organ damage. In this review, we highlight the current knowledge about the diagnosis and management of these complications during pregnancy.

## 1. Introduction

Thrombotic microangiopathies (TMAs) are a group of diseases characterized by the presence of microangiopathic hemolytic anemia (MAHA), thrombocytopenia, and microvascular thrombosis, and they are followed by organ damage of varying severity. Such forms may be acquired or inherited and acute or progressive, and they occur in both pediatric and adult populations. Prompt recognition and the initiation of early and specific therapy substantially reduce mortality [1]. Although TMAs may show different etiopathogenesis routes, they all show the presence of pathological lesions, which are characterized by endothelial damage and the formation of thrombi rich in platelets at the microvascular level, as a common denominator, and thrombotic damage to microcirculation pathways induces “mechanical” hemolytic anemia (microangiopathic), the consumption of platelets, and ischemic organ damage, all of which particularly affect the central nervous system and kidneys [2]. The primary forms of TMAs, i.e., those from ADAMTS13 (a disintegrin and metalloprotease with thrombospondin type 1 domain 13) deficiencies and those mediated by complements, may be either genetic or acquired, whereas the metabolism- and coagulation-mediated forms may be exclusively genetic. The secondary forms are, in general, associated with drug intake (DITMAs, or drug-induced TMAs), systemic infections (such as STEC-HUS, a TMA triggered by toxic infection by Shiga toxin), neoplastic diseases, pregnancy, conditions related to malignant hypertension, autoimmune diseases (systemic lupus erythematosus, systemic sclerosis, or antiphospholipid antibody syndrome), and the transplantation of solid organs or marrow. Pregnancy-related TMAs are rare, with an incidence of 1:25,000 births [3], accounting for 8% to 18% of all cases of TMAs. The physiology of pregnancy can promote the onset or recurrence of a TMA, particularly in the second and third trimesters, through the following two mechanisms: a progressive increase in von Willebrand factor during gestation and an increase in and accumulation of ultra-large multimers, and/or a physiological reduction in ADAMTS13 activity. These mechanisms are unable to counteract the possible presence of inhibitory antibodies. Cases of genetic ADAMTS13 deficiencies have also been recognized. The most common causes of mechanical hemolytic anemia and thrombocytopenia with organ involvement in pregnancy are preeclampsia/eclampsia and hemolysis, elevated liver enzymes, and low platelets (HELLP) syndrome, which are parts of the same syndrome, though with different presentations and levels of severity [4]; however, only HELLP syndrome is associated with microangiopathic hemolysis. The rarer forms of pregnancy-associated TMAs are hemolytic uremic syndrome (HUS) and thrombotic thrombocytopenic purpura (TTP). This review aimed to compare the diagnostic aspects of pregnancy-specific microangiopathies with the pathologic manifestations of other thrombotic microangiopathies.

## 2. Materials and Methods

During the preparation stage of this review, we searched PubMed using the following terms: “thrombotic microangiopathy”, “pregnancy”, “HELLP”, “HUS”, “TTP”, “preeclampsia”, and “hypertension”. We also identified the relevant literature via ‘snowballing’/citation-chasing, which was relevant for the background information. All studies were assessed based on the quality of their reporting, their bias in participant selection, the presentation of their results, and their authors’ conflicts of interest.

## 3. HELLP Syndrome

The acronym HELLP was coined in 1982 [5] to describe a syndrome characterized by hemolysis, elevated liver enzymes, and a reduced platelet count. Since that time, numerous reports have been published in the literature about a syndrome related to preeclampsia, though they are without uniformity in their diagnoses and characterizations. HELLP syndrome is a condition that complicates 0.1–0.2% of pregnancies, and 10–20% of pregnancies are complicated by severe preeclampsia [6,7]. The women at risk are older, white, and multiparous [8]. Characteristic of the third trimester of pregnancy, HELLP syndrome can also occur in the second trimester, with an estimated incidence of 11%. In 69% of cases, it occurs in the antepartum period, and in 31% of cases, it occurs in the postpartum period. In the latter cases, the typical onset is within 48 h of delivery [9].

### 3.1. Pathogenesis

Preeclampsia (PE) is defined as gestational hypertension (systolic blood pressure ≥ 140 mm Hg and/or diastolic blood pressure ≥ 90 mm Hg) accompanied by ≥1 of the following new-onset conditions at ≥20 weeks of gestation: proteinuria, acute kidney injury (a serum creatinine level ≥ 90 μmol/L), increased hepatic transaminases, a platelet count < 100 g/L, disseminated intravascular coagulation, hemolysis, fetal growth restriction, an abnormal umbilical artery Doppler wave form analysis, or stillbirth. Eclampsia, in addition to the criteria defining preeclampsia, is characterized by altered mental status, blindness, stroke, clonus, severe headache, and persistent visual scotomata. Although HELLP syndrome arises in women with hypertension and proteinuria, in 10–20% of cases, these signs are absent [4]. However, it must be considered part of the pathophysiologic spectrum of preeclampsia; therefore, where the signs and symptoms of the syndrome arise, the signs and symptoms of preeclampsia should also be sought. Excessive weight gain and generalized edema precede approximately 50% of the manifestations of the syndrome. As with preeclampsia, the pathogenetic origin lies in abnormal placentation, but compared with preeclampsia there is an excessive hepatic inflammatory response and excessive coagulatory activation [10]. Elevated circulating levels of the trophoblast-originated anti-angiogenic mediators, soluble endoglin, and the soluble form of the vascular endothelial growth factor receptor (sFlt1), which are the mediators of endothelial dysfunction, have been detected in PE/HELLP syndrome. Placental-derived sFlt1 binds to circulating vascular endothelial growth factor (VEGF) and placental growth factor (PlGF) and prevents their binding to innate receptors on endothelial cells [11]. The disrupted endothelial VEGF signaling results in endothelial cell dysfunction, glomerular endothelial swelling (endotheliosis), and subsequent hypertension and proteinuria. Disseminated intravascular coagulation (DIC) has often been reported in the setting of HELLP syndrome, especially in the context of postpartum hemorrhage, placental abruption, or fetal death [12,13].

### 3.2. Diagnosis

The typical symptoms of HELLP syndrome are epigastric or right upper abdominal quadrant pain, nausea, and vomiting. Serious abdominal pain may be fluctuating and colicky: many patients complain of malaise for days before clinical presentation, which is sometimes described to be similar to a viral syndrome. Approximately 30–60% experience migraines, and approximately 20% experience visual disturbances (scotomata) [4]. The symptoms of HELLP syndrome tend to be more severe in the nocturnal hours after a hospitalization that occurred in the daytime hours [14].

The laboratory diagnostic criteria of HELLP syndrome are summarized in Table 1 [15].

Hemolysis is the most notable feature of HELLP syndrome and is basically microangiopathic hemolytic anemia. The fragmentation of erythrocytes due to passage into a damaged endothelium is thus related to the extent of endothelial damage and the consequent extent of fibrin deposits (not only in the placenta). The presence of abnormal erythrocytes (schistocytes and echinocytes) on a peripheral smear also confirms a state of microangiopathic hemolysis [16]. An increase in the reticulocyte level reflects a compensatory release of immature erythrocytes. The destruction of erythrocytes causes an increase in the plasma level of lactate dehydrogenase (LDH) and a decrease in the hemoglobin level. Hemoglobinemia and macroscopic hemoglobinuria are found in 10% of cases [17]. The liberated hemoglobin is converted by the spleen into indirect bilirubin and can be bound in the circulation by haptoglobin. Moreover, the hemoglobin–haptoglobin complex, which is easily metabolized by the liver and liberated, lowers the plasma levels of this protein to the point where they may even be undetectable. Low levels of haptoglobin (<1 g/L–<0.4 g/L) can be used as a marker of hemolysis. Increased hepatic transaminases (AST and ALT) are directly related to liver damage resulting from microangiopathic hemolytic syndrome; thrombocytopenia is due to platelet activation and their relative adhesion to a damaged endothelium [18].

The Mississippi Triple-Class System classification [19] provides three classes based on the severity of thrombocytopenia (Table 2).

The variants of HELLP syndrome are classified on this basis:EL (elevated liver enzymes);HEL (hemolysis and elevated liver enzymes);ELLP (elevated liver enzymes and low platelet count);LP (low platelet count).

Women with partial HELLP syndrome also present with less symptomatology and develop fewer complications, although evolution to the full form is possible [18].

### 3.3. Complications

HELLP syndrome is associated with a 1% risk of maternal mortality and a variable risk of maternal morbidity, depending on the type of complication [20]:Disseminated intravascular coagulation (DIC) (15%);Untimely placental abruption (9%);Pulmonary edema (8%);Acute renal failure (3%);Liver failure and hemorrhage (1%);Adult respiratory distress syndrome (<1%);Sepsis (<1%);Stroke (<1%).

Headaches, visual disturbances, epigastric pain, and nausea/vomiting are better predictors of life-threatening complications than worsening laboratory parameters [21]. The spontaneous rupture of a subcapsular spathecular hematoma is rare in pregnancy but can be maternally life-threatening in 1 in 40,000/250,000 pregnancies, and approximately 1–2% of these cases occur in the presence of HELLP syndrome. Rupture most often occurs in the right hepatic lobe, and the symptoms are pain in the right upper quadrant of the abdomen radiating to the back, pain in the right shoulder, anemia, and hypotension. This event can be diagnosed via ultrasound, CT scan, or MRI [22]. Perinatal mortality and morbidity are considerably high and mainly depend on the time of gestation of the event. The perinatal mortality rate is estimated to be between 7.4% and 34% [23,24]. The main causes of mortality are prematurity, placental insufficiency with or without fetal growth retardation, and placental abruption [16]. With liver rupture, mortality can reach 80% [25]. Neonatal thrombocytopenia occurs in 15–38% of cases and is an important risk factor for neonatal intraventricular hemorrhage as well as long-term neurologic outcomes [26].

### 3.4. Management

Patients who are suspected to have HELLP syndrome should be hospitalized and observed with semi-intensive monitoring, given the typical rapidly progressing nature of the condition.

Initial management includes the following:Transfer to a Level II center (a center with multispecialty care and an adult/neonatal intensive care unit);Eclamptic seizure prophylaxis with magnesium sulfate for seizure prevention (any gestational age) and for fetal neuroprotection (24 to 32 weeks);Antihypertensives if blood pressure > 160/110 mmHg [27];Laboratory sampling every 12 h: CBC with platelet count, peripheral blood smear (if available), liver enzymes (AST/ALT), LDH, serum bilirubin, creatinine, blood glucose, and haptoglobin if available.

### 3.5. Delivery

The clinical course of HELLP syndrome is often characterized by a progressive deterioration in the maternal and fetal conditions with increased risk of morbidity and mortality, so a general consensus has been reached about the performance of delivery, at any gestational age and after the stabilization of maternal conditions, in the following situations:Gestational age < neonatal survival or ≥34 + 0 weeks;Fetal death in utero;Nonreassuring fetal tests (cardiotocography, biophysical profile, and Doppler velocimetry);

Severe maternal complications include the following: multiorgan dysfunction, DIC, hepatic infarction or hemorrhage, pulmonary edema, renal failure, or untimely placental abruption.

At gestational ages between the time of neonatal survival and 33 + 6 weeks, postponement of delivery for 24–48 weeks is indicated, if clinical conditions permit, to allow for the completion of the corticosteroid course for fetal maturation [28]. Vaginal delivery is preferred for women in labor, with ruptured membranes, or with a fetus in cephalic presentation at any gestational age. Labor can be induced in the case of a favorable cervix at a gestational age >30–32 weeks. A Caesarean section is to be performed for obstetrical indications; however, some authors indicate the need for a section at a gestational age < 30–32 weeks, especially if signs of fetal compromise are present (IUGR, oligohydramnios, and altered Doppler velocimetry). Given the increased risk of subfascial hematomas, some authors recommend the placement of a pelvic drain for 48 h following a Caesarean section [15]. The general agreement is that prophylactic platelet transfusion is not necessary for values > 50,000/mm^3^ in the absence of clinical bleeding or platelet dysfunction [29,30]. A platelet transfusion is indicated for values < 20,000/mm^3^ to avoid excessive bleeding in the case of vaginal delivery. In the case of a Caesarean section, some authors recommend platelet transfusion until a minimum level of 50,000/mm^3^ is reached [4]. The American Society of Anesthesiology does not recommend a platelet count limit regarding the safety of performing peripheral anesthesia [31]; however, a review of the literature indicates that with a minimum platelet count of 80,000/mm^3^, performing peripheral anesthesia and peridural catheter removal would be safe [32].

### 3.6. Expectant Management

In cases of HELLP syndrome with maternal and/or fetal complications, there is a general consensus regarding delivery at every gestational age. However, a broad consensus has not been reached for the management of the syndrome before 32 weeks in the presence of stable maternal conditions with moderately abnormal clinical parameters. In these cases, some authors have proposed a wait-and-see approach involving the use of bed rest, antihypertensives, permanent magnesium sulfate administration, antithrombotic agents (low-dose aspirin and dipyridamole), plasma expanders (crystalloids, albumin, and fresh frozen plasma), and steroids (prednisone, dexamethasone, and betamethasone) [33,34,35,36,37]. The results of these studies have indicated that the wait-and-see management of HELLP syndrome is possible in selected groups of patients < 32 weeks, and that with prolonged pregnancy, the perinatal outcomes are no different from those of cases of comparable gestational age in which delivery is accomplished in 48 h. Moreover, the latency time from delivery is not significantly increased. The available evidence indicates that the use of corticosteroids for the induction of fetal maturation improves the perinatal outcomes in women with HELLP syndrome before 34 weeks and that this therapy leads to an increase in the platelet count, a determinant factor in particular for the induction of epidural anesthesia, if needed [38]. The use of high-dose dexamethasone (10 mg iv every 6–12 h for two doses, followed by 5–6 mg every 6–12 h for two additional doses) is effective in improving clinical parameters in the pre- and postpartum periods [39]. However, evidence that the use of high-dose corticosteroids leads to improvements in maternal–fetal mortality and morbidity is lacking [40]. At present, a review by Cochrane [41] concludes that there is no evidence of improved maternal–fetal outcomes with corticosteroid use and that the only clinically demonstrable benefit is increased platelet counts.

### 3.7. Postpartum Management

After delivery, patients with HELLP syndrome should be monitored for vital parameters, fluid balance, and laboratory parameters for at least 48 h; however, in cases of severe renal, coagulation, or hemorrhagic complications, the postpartum period may coincide with a further worsening of clinical conditions. These patients are at risk of developing pulmonary edema from blood and blood product transfusions or incongruous fluid infusions in the presence of impaired renal function. In addition, there is a risk of acute renal failure and a need for dialysis [42], so these patients may require intensive monitoring and treatment for several days. The criteria for admission to an intensive care unit are the occurrence of sepsis, pulmonary edema, hypertension unresponsive to drugs, anuria, repeated seizure episodes, massive blood loss with DIC, neurological dysfunction requiring ventilatory assistance (cerebral hemorrhage or cerebral edema), and critical abdominal disease (acute fatty liver hepatic rupture, ruptured arterial aneurysm, or adrenal hemorrhage) [43]. The first onset of HELLP syndrome, as mentioned above, can occur after delivery, in a period that can vary from a few hours to 7 days. Therefore, all expectant mothers and midwives should be educated to recognize the early signs of this syndrome. The treatment of these patients is similar to that during pregnancy, including the prophylaxis of seizures with magnesium sulfate.

### 3.8. Counseling and Recurrence

Women with HELLP syndrome are at increased risk of developing all forms of preeclampsia in a subsequent pregnancy. In general, this risk stands at 20%; however, it is higher during second-trimester HELLP syndrome. The results of a meta-analysis of 512 women with HELLP syndrome showed a recurrence rate in subsequent pregnancies of 7%, a rate of 18% for subsequent preeclampsia, and a rate of 18% for subsequent gestational hypertension [44]. Epidemiological studies have demonstrated an association between preeclampsia/HELLP syndrome and the future development of cardiovascular disease. A meta-analysis that targeted women with a previous history of severe preeclampsia/HELLP syndrome indicated doubled risks of heart disease, cerebrovascular accidents, and early peripheral arterial disease [45].

## 4. Hemolytic Uremic Syndrome (HUS)

Hemolytic uremic syndrome (HUS) is characterized by the triad of microangiopathic hemolytic anemia, thrombocytopenia, and acute kidney injury. It is a rare disorder. Its estimated incidence among the general population is 0.5 in 10,000,000 people and it accounts for around 16–20% of women with TMAs. It most frequently occurs late in the third trimester or in the postpartum period. Atypical hemolytic uremic syndrome (aHUS) is a complement-mediated disorder characterized by microangiopathic hemolysis, thrombocytopenia, and renal failure. It should be distinguished from typical diarrhea-associated hemolytic uremic syndrome, which is most commonly due to Shiga toxin-producing *Escherichia coli*. Approximately 10–20% of aHUS diagnoses occur in the setting of pregnancy, where it has been termed pregnancy-associated atypical hemolytic uremic syndrome. Pregnancy is a complement-amplifying condition, and maternal exposure to semi-allogenic fetoplacental material increases during gestation, with peak exposure at delivery [46,47]. Historically, HUS was defined as not being associated with diarrhea, but the usefulness of this classification was limited because aHUS can present with gastrointestinal symptoms such as nausea, vomiting, and diarrhea. However, the presence of diarrhea is an important symptom and can trigger further evaluation. This is especially true when the diarrhea is hemorrhagic with concomitant microangiopathic hemolytic anemia and thrombocytopenia. In such cases, the most likely etiology is Shiga toxin produced by *Escherichia coli (E. coli*), and the condition is called STEC-HUS. This diagnosis can be confirmed via a Shiga toxin immunoassay on a stool sample and tested via polymerase chain reaction (PCR) or cultures [48]. The primary treatment for STEC-HUS is supportive care.

aHUS is characterized by microangiopathic hemolytic anemia, thrombocytopenia, and organ damage. Acute kidney damage is almost always present, and an elevated serum creatinine level is used as an entry criterion for clinical trials involving subjects with aHUS [49]. The number of these patients progressing to end-stage kidney disease (ESKD) is approximately 10–12% under terminal complement blockade [50]. Because of the inflammatory nature of aHUS, additional signs and symptoms are often present, such as neurological symptoms (headaches, confusion, and seizures) or gastrointestinal symptoms (abdominal pain and diarrhea) [51]. Uncontrolled complement activation is, to date, considered the primary cause of aHUS. Mutations in genes regulating the complement system have been reported in more than 60% of aHUS cases [52]. These can be mutations with loss of function or gain of function in complement activators. Genetic mutation analysis is used to predict the risk of relapse in many cases of aHUS, but in others, the outcome is unfavorable despite the identification of the complement abnormality [51]. When aHUS arises from secondary causes, the primary treatment is directed at the etiology. When the primary etiology cannot be easily treated or when aHUS is refractory, a complement blocker treatment with eculizumab should be considered. Eculizumab was approved by the FDA as a treatment for aHUS in 2011 and has substantially improved the long-term prognosis of the disease. When aHUS arises in pregnancy or the postpartum period, it is referred to as pregnancy-associated atypical hemolytic uremic syndrome (p-aHUS). According to the literature, p-aHUS often arises in the context of obstetric complications, such as preeclampsia, placental abruption, fetal death, or postpartum hemorrhage [48,53]. However, it can occur even after uncomplicated deliveries. Because its features overlap with HELLP syndrome, p-aHUS should be considered in patients with microangiopathic hemolytic anemia and thrombocytopenia if they are at <20 weeks of gestation or >48–72 h postpartum or if there is a family or personal history of HUS. For patients beyond 20 weeks and before 48 h postpartum, diagnosis can be difficult. The differentiation between HELLP and p-aHUS is crucial because the first-line treatment for p-aHUS is a complement blockade with eculizumab [54,55]. Because of delayed diagnosis, corticosteroids and plasmapheresis are often attempted as first-line agents [56]. These treatments have minor effects on p-aHUS, and their use may increase maternal risk and delay appropriate care.

## 5. Thrombotic Thrombocytopenic Purpura (TTP)

Thrombotic thrombocytopenic purpura is characterized by microangiopathic hemolytic anemia with severe thrombocytopenia and variable organ ischemia, most commonly neurologic, cardiac, or renal [57]. TTP is most often caused by the acquisition of inhibitory autoantibodies against ADAMTS13 [58]. Rarely, congenital TTP (Upshaw–Schulman syndrome) arises from a mutation of the ADAMTS13 gene [59]. In the past, the diagnosis was only clinical, but with the advent of ADAMTS13 assays and the possibility of sequencing the ADAMTS13 gene, the diagnosis of both autoimmune and congenital TTP can be quickly and efficiently confirmed [60]. A genetic ADAMTS13 deficiency presents 100% of the TTP risk during pregnancy, so adult-onset constitutive TTP can be encountered for the first time during pregnancy with an “HELLP” presentation [61]. During pregnancy, prompt recognition and differentiation from preeclampsia or HELLP syndrome, followed by appropriate treatment, are critical, as maternal–fetal morbidity and mortality are high if unrecognized [62]. In pregnant patients with immune-mediated TTP, the acute phase should be managed with plasma exchange with the addition of corticosteroids if tolerated [62]. During remission after an episode during pregnancy, congenital TTP patients may require prophylactic therapy prior to and during their next pregnancy. The recently published International Society on Thrombosis and Haemostasis (ISTH) guidelines for the management of TTP state that pregnant congenital TTP patients should receive prophylactic plasma infusions to prevent relapse [63].

Particular caution should be observed for patients with TTP in the setting of a severe ADAMTS13 deficiency regarding platelet transfusions, as they can cause thrombotic complications and/or a worsening of neurological symptoms. When a platelet transfusion becomes absolutely necessary to prepare for obstetric procedures (e.g., Caesarean section or central catheter insertion), this should be performed with concomitant fresh frozen plasma administration (perfusion or plasma exchange) [58].

## 6. Differential Diagnosis

The borders between the different forms and presentations of TMAs are not well defined, so differential diagnosis may be difficult or even impossible because these conditions may coexist. To further complicate the diagnostic process, during pregnancy, the hematologic, proteinuria, and complement parameters have different reference ranges than in nonpregnant patients [64]. The differential diagnosis between these diseases requires a careful clinical evaluation and, especially, an assessment of laboratory parameters. In particular, TTP and aHUS associated with pregnancy or the postpartum period require urgent management, biological testing, and a specific treatment. A biochemical index much studied in recent years, the soluble fms-like tyrosine kinase-1/placental growth factor (sFlt1/PlGF) ratio, may be a valuable tool for initial assessment in patients with a TMA in pregnancy to rule out preeclampsia/HELLP syndrome. sFlt-1/PlGF ratios above 85 before 34 weeks of gestation and above 110 after 34 weeks seem to be strongly suggestive of hypertensive disorders of pregnancy [65,66]. This type of determination may not always be available in all centers, and further studies are needed to confirm the diagnostic value of the sFlt-1/PlGF ratio in pregnancy-associated TMAs. Screening for hemolysis with the LDH dosage is not routinely performed in pregnancy, and testing modalities vary among laboratories. However, it is the most-studied parameter in HELLP syndrome, aHUS, and TTP, but other options include total or indirect bilirubin, haptoglobin, and peripheral blood smears. LDH values >1000 U/L are common in aHUS but are less frequent in HELLP syndrome. Factors that may favor the diagnosis of aHUS include a personal or family history of aHUS and reduced hemoglobin (<8 g/dL). A high LDH/AST ratio suggests that red blood cell hemolysis is disproportionate to hepatic inflammation. The hemoglobin level is typically very low in aHUS but may be normal (>11 g/dL) in HELLP syndrome due to hemoconcentration. If the LDH level is elevated, other tests should be used to confirm microangiopathic hemolytic anemia, including peripheral blood smear, haptoglobin, and/or direct Coombs tests. By definition, AST and ALT levels are always elevated in HELLP syndrome. An elevation > 2 times the normal value is also characteristic of complicated preeclampsia. Although aHUS is not classically associated with increased liver enzymes, numerous reports on pregnancy-associated aHUS have described increases in AST and ALT levels similar to those in HELLP syndrome. Therefore, increased AST or ALT levels do not rule out TTP or aHUS. A reduction in the platelet count is common in HELLP syndrome, TTP, and aHUS. However, profound thrombocytopenia should increase suspicion for TTP, particularly when the platelet count is <30,000/mm^3^ [64]. In some cases with obstetric indications, it may be a priority to interrupt the pregnancy for the birth of the fetus. It could be reasonable to assess ADAMTS13 for TTP when the platelet count is <70,000/mm^3^. Some platelet count abnormalities in the absence of another etiology, as in idiopathic thrombocytopenic purpura, can be explained by gestational thrombocytopenia. In addition, a decrease in platelet count >25% from normal values, regardless of the absolute platelet count, defers an evaluation for microangiopathic hemolytic anemia. To exclude the diagnosis of TTP, no parameter is sufficiently discriminating, and the only reliable biological test is the ADAMTS13 assay. When TTP is suspected and a vital organ (heart or nervous system) is involved, in the presence of a low platelet count, plasma exchange should be considered until the ADAMTS13 result is available. In such cases, the sample for the ADAM13 analysis should be taken before the plasma infusion.

In patients with hemolytic anemia and a decreased platelet count but normal ADAMTS13 activity, the differential diagnosis between HELLP and aHUS remains challenging. The platelet count is not sufficient to distinguish between these diseases, so other parameters should be evaluated. A marked increase in serum creatinine levels (>2 times normal values) is uncommon in patients with HELLP syndrome [67]. Although increases >1.1 mg/dL are relatively common [64], elevations > 2.0 mg/dL in pregnant or postpartum patients should direct evaluations for aHUS. Specifically, microangiopathic hemolytic anemia should be evaluated. This evaluation should be performed regardless of the platelet count or liver enzymes. In addition, a persistent increase in serum creatinine > 1.1 mg/dL > 72 h postpartum should direct the evaluation toward aHUS. Delivery is the definitive treatment for HELLP syndrome [68]. In many cases, improvements in clinical signs occur within 48–72 h of delivery [58]. A prolongation of abnormalities on laboratory tests may persist in patients with obstetric complications such as placental abruption, postpartum hemorrhage, or sepsis. In the absence of complications at delivery, the persistence of abnormalities on examinations more than 72 h postpartum should direct suspicion toward aHUS. When hemolysis and renal damage persist or worsen after delivery without another explanation, the diagnosis is probably aHUS. A rapid and dramatic increase in the serum creatinine level, often >3–4 mg/dL, is characteristic of aHUS; in these cases, the complement blockade should be accelerated [69]. In cases with normal ADAMTS13 activity, plasma exchange should not be considered as a first-line treatment.

The clinical features of TMAs in pregnancy, thrombocytopenia, MAHA, and organ injury, may overlap with other conditions. In particular, there are autoimmune diseases that can result in TMAs such as antiphospholipid antibody syndrome (APS), a systemic autoimmune disorder characterized by arterial or venous thromboses, vasculopathy, and obstetric events that occur in the presence of antiphospholipid (APL) antibodies [70]. Catastrophic antiphospholipid syndrome (CAPS) is a rare and potentially fatal form of APS characterized by severe thrombotic complications occurring in multiple organs over a short period of time or simultaneously. APS can occur as a primary disorder or in association with an underlying autoimmune disease, such as systemic lupus erythematosus (SLE). For APS, the presence of antiphospholipid (APL) antibodies is diagnostic (lupus anticoagulant, anticardiolipin antibodies, and anti-β2 glycoprotein-I). TMAs should be considered in patients with SLE and new-onset acute kidney injury or malignant hypertension with or without signs of nephritis. Genetic and functional complement testing should be considered in patients with TMAs associated with SLE or APS. A kidney biopsy should be pursued after delivery to establish a diagnosis and initiate an appropriate treatment. MAHA and thrombocytopenia are also present in 50% of cases of scleroderma renal crisis (SRC) [71], a rare but life-threatening complication of systemic sclerosis (SSc). The increased activation of both the classic and alternative complement pathways has been demonstrated via plasma biomarkers [72] and through immunohistochemical staining of kidney biopsy specimens. A rare pregnancy-specific condition that may present with some TMA features is Acute Fatty Liver of Pregnancy (AFLP), a rare and life-threatening condition associated with coagulopathy, liver failure, and multiorgan involvement [73]. This condition appears to be the result of mitochondrial dysfunction during the oxidation process of fatty acids. AFLP occurs during the third trimester with nonspecific symptoms of malaise, anorexia, nausea, and vomiting. Usually, there is no associated hypertension or proteinuria, but metabolic acidosis, acute liver failure, and low-grade DIC are present. Delivery is the definitive treatment for AFLP. Laboratory abnormalities can include hypoglycemia and elevated serum bilirubin, aspartate aminotransferase (AST), and alanine aminotransferase (ALT). Elevated ammonia and an elevated white blood cell count may be present. Low platelets and coagulopathy may be present, which manifest as a prolonged prothrombin time and international normalized ratio and low fibrinogen. Imaging with a CT scan, preferably after delivery, may reveal a steatotic liver, even if the macroscopic steatosis aspect of the liver may be checked by the obstetrician during a Caesarian section. Testing for the enzyme long-chain 3-hydroxyacyl-CoA dehydrogenase (LCHAD) is recommended because some babies from women with AFLP carry the metabolic disease of beta oxidation, and testing the baby is preferred; if not possible, the mother is tested, as she is potentially heterozygous. Finally, the term pseudo-TMA describes TMAs secondary to cobalamin (vitamin B12) deficiency. A deficiency of vitamin B12 causes a syndrome with pancytopenia, megaloblastic anemia, hypersegmented neutrophils (explained by dysfunctional intramedullary hematopoiesis), and dorsal column dysfunction (explained by dysfunctional myelin synthesis). Cobalamin deficiency resulting in hemolysis is very rare and is seen in only 10% of deficiencies. Of those, cobalamin deficiency causing a pseudo-TMA is even rarer and is seen in only about 2.5% of cases [74]. Patients with a pseudo-TMA present with hemolytic anemia, thrombocytopenia, and dysmorphic “fragmented” red blood cells (RBCs). They are often misdiagnosed with other TMA syndromes and receive unnecessary therapies. The treatment for vitamin B12 deficiency-induced hemolysis is simply the parenteral replacement of vitamin B12. Even if the most common neurologic findings include myelopathy, peripheral neuropathy, optic neuropathy, altered mental status, and dementia, the presence of such deficits is only seen in about 40% of cases of cobalamin deficiency. Studying the hematological picture can lead to the exact identification of a vitamin deficiency. The reticulocyte count is low, given the unavailability of B12, for the expected compensatory increase in erythropoiesis. The presence of macrocytosis is another clue toward a pseudo-TMA. A high LDH level and reticulocyte production index (RPI) can be useful laboratory tools to differentiate pseudo-TMAs from other primary TMA syndromes such as TTP. Hyperhomocysteinemia is related to cobalamin deficiency since in the absence of vitamin B12 homocysteine cannot be methylated and converted into Methionine. The accumulation of homocysteine, given its pro-oxidant activity, could underlie hemolytic anemia due to red blood cell damage and microangiopathy due to systemic endothelial damage [75]. For homocysteine sampling, a fasting specimen is preferred to establish baseline values.

## 7. Conclusions

Thrombotic microangiopathies are a group of potentially very serious diseases with complex pathophysiological bases. In pregnancy, TMAs take on even more special connotations because of the biological changes during pregnancy itself, and the fetal and neonatal implications, and because of the therapeutic possibilities. During pregnancy, the manifestations of common TMAs intersect with similar pathologies that do not exist in the nonpregnant adult population. In this review, we wanted to focus on the diagnostic aspects during pregnancy by conducting a review of the various forms of TMAs with the newest terminologies established in the literature in recent years. The diagnostic aspects have been linked, in a reasoned manner, to the state of pregnancy, the time of pregnancy, the therapeutic possibilities allowed by the state of pregnancy, and the future of the mother and fetus.

## Figures and Tables

**Table 1 diagnostics-14-00352-t001:** Diagnostic criteria of HELLP syndrome.

Hemolysis	Increased Liver Enzymes	Thrombocytopenia
Peripheral smear abnormalities Total bilirubin ≥ 1.2 mg/dL (or ≥20.5 μmol/L) Lactate dehydrogenase (LDH) > 600 U/L	Aspartate aminotransferase (AST) and/or aspartate alaninotransferase (ALT) > 70 U/L	Platelet count < 100,000/mm^3^

**Table 2 diagnostics-14-00352-t002:** Mississippi Triple-Class System classification of HELLP syndrome.

Class 1	Class 2	Class 3
PLT ≤ 50,000/mm^3^ LDH ≥ 600 U/L AST and/or ALT ≥ 70 U/L	PLT > 50,000/mm^3^ ≤ 100,000/mm^3^ LDH ≥ 600 U/L AST and/or ALT ≥ 70 U/L	PLT > 100,000/mm^3^ ≤ 150,000/mm^3^ LDH ≥ 600 U/L AST and/or ALT ≥ 40 U/L.

## Data Availability

No new data were created or analyzed in this study. Data sharing is not applicable to this article.

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
