# Peer review of "HELLP Syndrome and Differential Diagnosis with Other Thrombotic Microangiopathies in Pregnancy"

_diagnostics, 2024, doi:10.3390/diagnostics14040352_

Round 1

Reviewer 1 Report

Comments and Suggestions for Authors

The subject is of importance and the manuscript is well written, however there are some major concerns for important messages to be delivered/paragraph 5

1/In differential diagnosis these diseases should appear in the last paragraph: auto immune diseases/catastrophic antiphospholipid syndrome, multiple sclerosis, acute fatty acid liver disease ; cobalamin deficiency. Biological tests are done if these diagnosis are suspected // anti nuclear antibody, anti ARN Pol 3,…Lupus anticoagulant markers, search for liver failure (hypoglycemia, hypofibrinogenemia, low prothrombin time, antithrombin 3..), homocystein and vitamin B9 and B12.

2/The only reliable biological test to rule out TTP is ADAMTS13 dosage and no routine analysis parameter is discriminant. So if the is a vital organ involved (cardiac, neurologic) with low platelet count and TTP suspected, plasma exchange should be considered until ADAMTS13 result with sample taken before any plasma infusion.

3/The ratio sFlt1/PLGF is of very high value to rule out HELLP/placental toxemia so this must appear also

Other minor concerns :

Page 2, Line 45 : Shigatoxin HUS is usely not classified in primary TMAs but rather in infection mediated TMA // PMID: 27868334. The abbreviation SEU D+ or ST-SEU is unusual and STEC-HUS should be preferred (as it has been used later in the manuscript)…

Page 6, Line 289 ; Page 8 Line 356 square brackets must be removed

Page 6, Line 295 diagnosis confirmed by /precise: stool sample

Page 7, Line 302 10-12% of ESKD/precise: under terminal complement blockade

Page 7, Line 308, reference is missed with empty brackets

Page 7, Line 334, an important information is to say that genetic ADAMTS13 deficiency presents 100% of TTP risk during pregnancy and so adult onset constitutive TTP can be encountered for the first time during pregnancy with a “HELLP” presentation. // Cite PMID: 22547583

Page 7 : Line 345, it would be of importance to give the information of platelet transfusion risk in the setting of severe ADAMTS13 deficiency and to do so if absolutely required with concomitant fresh frozen plasma administration (perfusion or plasma exchange)/cite PMID: 32808006

Page 8, Line 360, Ref 65 ; the validation of biologic parameters of LDH and LDH/AST ratio to differentiate TTP from HELLP has not been reproduced in other studies or case series. This either should be removed or at least a comment is required

Page 8, Line 391, Ref error ; Ref 59 can be cited

Author Response

COMMENT

In differential diagnosis these diseases should appear in the last paragraph: auto immune diseases/catastrophic antiphospholipid syndrome, multiple sclerosis, acute fatty acid liver disease ; cobalamin deficiency. Biological tests are done if these diagnosis are suspected // anti nuclear antibody, anti ARN Pol 3,…Lupus anticoagulant markers, search for liver failure (hypoglycemia, hypofibrinogenemia, low prothrombin time, antithrombin 3..), homocystein and vitamin B9 and B12.

RESPONSE

As suggested a dissertation on other causes of microangiopathy has been included in the "differential diagnosis" section.

COMMENT

The only reliable biological test to rule out TTP is ADAMTS13 dosage and no routine analysis parameter is discriminant. So if the is a vital organ involved (cardiac, neurologic) with low platelet count and TTP suspected, plasma exchange should be considered until ADAMTS13 result with sample taken before any plasma infusion.

RESPONSE

We thank the Reviewer for the valuable suggestion that was included in the text in “differential diagnosis” section.

COMMENT

The ratio sFlt1/PLGF is of very high value to rule out HELLP/placental toxemia so this must appear also.

RESPONSE

The sFlt-1/PlGF index has been included in the "differential diagnosis" section as a discriminative possibility between preeclampsia/HELLP syndrome and other thrombotic microangiopathies in pregnancy

COMMENT

Page 2, Line 45 : Shigatoxin HUS is usely not classified in primary TMAs but rather in infection mediated TMA // PMID: 27868334. The abbreviation SEU D+ or ST-SEU is unusual and STEC-HUS should be preferred (as it has been used later in the manuscript)…

RESPONSE

As suggested, the text has been modified and Sgigatoxin HUS has been included among secondary TMAs (the suggested reference is cited later in the text).

COMMENT

Page 6, Line 289 ; Page 8 Line 356 square brackets must be removed

RESPONSE

Square brackets have been removed.

COMMENT

Page 6, Line 295 diagnosis confirmed by /precise: stool sample

RESPONSE

Sample type was added for confirmation of diagnosis.

COMMENT

Page 7, Line 302 10-12% of ESKD/precise: under terminal complement blockade

RESPONSE

The sentence was completed as suggested

COMMENT

Page 7, Line 308, reference is missed with empty brackets

RESPONSE

The entire sentence was deleted as incorrect

COMMENT

Page 7, Line 334, an important information is to say that genetic ADAMTS13 deficiency presents 100% of TTP risk during pregnancy and so adult onset constitutive TTP can be encountered for the first time during pregnancy with a “HELLP” presentation. // Cite PMID: 22547583.

RESPONSE

We have taken up the valuable suggestion and transferred it to the text.

COMMENT

Page 7 : Line 345, it would be of importance to give the information of platelet transfusion risk in the setting of severe ADAMTS13 deficiency and to do so if absolutely required with concomitant fresh frozen plasma administration (perfusion or plasma exchange)/cite PMID: 32808006.

RESPONSE

As suggested, the text has been modified by including the risks of platelet transfusions in women with TTP and citing the reference.

COMMENT

Page 8, Line 360, Ref 65 ; the validation of biologic parameters of LDH and LDH/AST ratio to differentiate TTP from HELLP has not been reproduced in other studies or case series. This either should be removed or at least a comment is required.

RESPONSE

The data in this sentence have been removed as unconsolidated and irrelevant.

COMMENT

Page 8, Line 391, Ref error ; Ref 59 can be cited

RESPONSE

References have been corrected.

Reviewer 2 Report

Comments and Suggestions for Authors

Dear author’s,

I was pleased to review your article and i have the following comment’s:

First of all it is not known the type of your article. It is a review?

Why do you choose this topic and please highlight what new info brings your manuscript in the field.

The article should contain a section with the methodology - how this info was obtained.

Please add a short conclusion at the end in order to clarify the article aim and hypothesis.

Minor English edits.

Author Response

COMMENT

First of all it is not known the type of your article. It is a review?

RESPONSE

As suggested, the type and purpose of this article have been specified at the end of the “Introduction” section.

COMMENT

Why do you choose this topic and please highlight what new info brings your manuscript in the field.

Please add a short conclusion at the end in order to clarify the article aim and hypothesis.

RESPONSE

The new "conclusions" section included in the text specifies the purpose of a review on this topic with a focus on the latest terminology with a particular obstetric perspective.

COMMENT

The article should contain a section with the methodology - how this info was obtained.

RESPONSE

A new "Materials and Methods" section has been added, as suggested, to explain the research methodology and compilation of the review.

Round 2

Reviewer 1 Report

Comments and Suggestions for Authors

The manuscript has been markedly improved and is better organized.

Just a last comment line 452 453 "AFLP and diagnosis can be confirmed by testing for the enzyme long-chain 3-hydroxyacyl-CoA dehydrogenase."

This is not correct : AFLP is a syndrome diagnosed on clinical and biological parameters but overall, CT scan liver aspect of a steatosic liver (and sometimes macroscopic steatosis aspect of the liver by the obstetrician check during cesarian section) ; testing of LCHAD is recommended because some of the babies from women with a AFLP carry metabolic disease of beta oxydation and testing is preferred on the baby ; if not possible, the mother  is tested as potential heterozygous.

Author Response

REVIEWER 1

The manuscript has been markedly improved and is better organized.

COMMENT

Just a last comment line 452 453 "AFLP and diagnosis can be confirmed by testing for the enzyme long-chain 3-hydroxyacyl-CoA dehydrogenase."

This is not correct : AFLP is a syndrome diagnosed on clinical and biological parameters but overall, CT scan liver aspect of a steatosic liver (and sometimes macroscopic steatosis aspect of the liver by the obstetrician check during cesarian section) ; testing of LCHAD is recommended because some of the babies from women with a AFLP carry metabolic disease of beta oxydation and testing is preferred on the baby ; if not possible, the mother  is tested as potential heterozygous.

RESPONSE

We took the Reviewer's valuable advice and reformulated the diagnostic criteria of the AFLP.

Reviewer 2 Report

Comments and Suggestions for Authors

Thank you for your response.

Author Response

COMMENT

Thank you for your response.

RESPONSE

We thank the Reviewer for liking the latest version of the manuscript.